# Coronavirus Antiviral Research Database (CoV-RDB): An Online Database Designed to Facilitate Comparisons between Candidate Anti-Coronavirus Compounds

**DOI:** 10.3390/v12091006

**Published:** 2020-09-09

**Authors:** Philip L. Tzou, Kaiming Tao, Janin Nouhin, Soo-Yon Rhee, Benjamin D. Hu, Shruti Pai, Neil Parkin, Robert W. Shafer

**Affiliations:** 1Division of Infectious Diseases, Stanford University School of Medicine, Stanford, CA 94305, USA; kaiming.tao@stanford.edu (K.T.); jnouhin@stanford.edu (J.N.); syrhee@stanford.edu (S.-Y.R.); 2Undergraduate School of Humanities and Sciences, Stanford University, Stanford, CA 94305, USA; benhu8@stanford.edu; 3Undergraduate Studies, University of California, Berkeley, CA 94720, USA; shrutipai@berkeley.edu; 4Data First Consulting Inc., Sebastopol, CA 95472, USA; nparkin34@gmail.com

**Keywords:** coronavirus, COVID-19, SARS-CoV-2, SARS-CoV, MERS-CoV, antiviral therapy

## Abstract

Background: To prioritize the development of antiviral compounds, it is necessary to compare their relative preclinical activity and clinical efficacy. Methods: We reviewed in vitro, animal model, and clinical studies of candidate anti-coronavirus compounds and placed extracted data in an online relational database. Results: As of August 2020, the Coronavirus Antiviral Research Database (CoV-RDB; covdb.stanford.edu) contained over 2800 cell culture, entry assay, and biochemical experiments, 259 animal model studies, and 73 clinical studies from over 400 published papers. SARS-CoV-2, SARS-CoV, and MERS-CoV account for 85% of the data. Approximately 75% of experiments involved compounds with known or likely mechanisms of action, including monoclonal antibodies and receptor binding inhibitors (21%), viral protease inhibitors (17%), miscellaneous host-acting inhibitors (10%), polymerase inhibitors (9%), interferons (7%), fusion inhibitors (5%), and host protease inhibitors (5%). Of 975 compounds with known or likely mechanism, 135 (14%) are licensed in the U.S. for other indications, 197 (20%) are licensed outside the U.S. or are in human trials, and 595 (61%) are pre-clinical investigational compounds. Conclusion: CoV-RDB facilitates comparisons between different candidate antiviral compounds, thereby helping scientists, clinical investigators, public health officials, and funding agencies prioritize the most promising compounds and repurposed drugs for further development.

## 1. Introduction

The Coronavirus Antiviral Research Database (CoV-RDB) is designed to promote uniform reporting of experimental results; to facilitate comparisons between different candidate antiviral compounds; and to help scientists, clinical investigators, public health officials, and funding agencies prioritize the most promising compounds and repurposed drugs for further development. By comprehensively reviewing published laboratory, animal model, and clinical data on potential coronavirus therapies, CoV-RDB also makes it unlikely that promising treatment approaches will be overlooked.

## 2. Methods/Results

CoV-RDB contains four main types of antiviral experimental data, six main lookup/explanation tables, and a registry of ongoing or planned clinical trials. The four main types of antiviral experimental data include (i) cell culture and entry assay experiments; (ii) biochemical experiments; (iii) animal model studies; and (iv) clinical studies. The six main lookup/explanation tables provide information on viruses, virus strains/isolates, tested compounds, compound targets, cell types, and animal models.

CoV-RDB data are stored in a PostgreSQL relational database, but there is not necessarily a one-to-one relationship for the tables displayed on the web and their underlying database structure. Indeed, several of the website tables contain information from more than one underlying database table. As of 14 August 2020, the CoV-RDB contains data from more than 1800 virus cell culture experiments, 465 entry assay experiments, 519 biochemical experiments, 259 animal model experiments, and 71 clinical studies from more than 310 peer-reviewed publications and 90 preprints.

Research articles are identified through incremental daily searches of PubMed and bioRxiv using the search term “coronavirus” and by citations identified through reading these papers. Publications containing experimental data are imported into a staging Zotero database and then annotated to extract the data described in the sections below. Preprints that are subsequently published in a peer-reviewed journal are identfied using a computer script that parses datasets downloaded from the Stephen B. Thacker CDC (Centers for Disease Control and Prevention) Library [1]. Clinical trial datasets are retrieved from clinicaltrials.gov and WHO (World Health Organization) ICTRP (International Clinical Trials Registry Platform). Clinical trials are linked to corresponding articles when they are published (Figure 1). The following sections describe the purpose and contents of the CoV-RDB tables displayed on the web.

### 2.1. Experimental Data Tables

#### 2.1.1. Cell Culture and Entry Assay Experiments

The cell culture experiments table contains 13 fields, including four fields present in each of the experiment tables: reference, compound, virus category, and virus isolate/strain. The nine fields unique to the cell culture experiments table include six that describe experimental conditions and three that contain experimental results. The six experimental conditions include the (i) cells used for antiviral testing, (ii) multiplicity of infection (MOI; the virus titer divided by the number of cells), (iii) time between addition of drug and addition of virus, (iv) drug concentration(s), (v) duration of virus infection, and (vi) indicator of virus replication.

The three experiment results are the half-maximal effective concentration (EC_50_), percent inhibition, and the 50% cytotoxic concentration (CC_50_). The EC_50_ can only be determined using a series of compound dilutions. While the EC_50_ is usually reported as μM, inhibitory activity for interferons is also often reported as international units (IU)/mL and inhibitory activity for monoclonal antibodies is often reported as ng/mL. The EC_50_ is available for the vast majority of in vitro cell culture experiments. However, for a few experiments, the experimental setup involved a single compound concentration (rather than a dilution series). For these experiments, the percent virus inhibition with the single compound concentration is reported.

There are two tables for entry assay experiments—one for pseudovirus entry assays and another for cell–cell fusion assays. The pseudovirus assay table contains the following six unique fields: (i) pseudovirus vector, (ii) pseudovirus number, (iii) target cell type, (iv) time to addition of drug, (v) indicator of virus replication, and (vi) EC_50_. In the pseudovirus experiments, the virus strain is a virus construct composed of a virus that does not require a Biosafety Level 3 (BSL-3) laboratory, such as vesicular stomatitis virus (VSV) or human immunodeficiency virus type 1 (HIV-1), into which the coronavirus *spike* (*S*) gene has been cloned. This construct also has a reporter gene such as luciferase or GFP. The cell–cell fusion assay table contains the following seven unique fields: (i) effector cell type, (ii) effector cell number, (iii) target cell type, (iv) target cell number, (v) time to addition of drug, (vi) indicator of virus replication, and (vii) EC_50_.

#### 2.1.2. Biochemical Experiments

The biochemical experiments table contains two unique fields: the biochemical target and the half maximal inhibitory concentration (IC_50_). The biochemical target is usually one of the virus enzymes including RNA-dependent RNA polymerase (RdRP), main protease (also called 3C-like protease; 3CL^pro^), papain-like protease (PL^pro^), and helicase. However, cell-free assays that test inhibitors of the spike (S) protein binding to angiotensin converting enzyme 2 (ACE2) are also included.

#### 2.1.3. Animal Model Studies

The animal model experiments are characterized by comparisons between a group of animals receiving a treatment intervention either shortly before or after virus infection and a group of untreated virus-infected control animals. The animal model experiments table has two parts—experimental conditions and experimental results. The experimental conditions include the (i) animal model, (ii) size and route of virus challenge, (iii) treatment intervention, (iv) treatment dosage, (v) treatment timing in relation to the addition of virus, (vi) number of treated subjects, and (vii) number of control subjects. The experimental results, which often depend on the study, include endpoints such as mortality, weight loss, fever, respiratory rate, lung pathology, and virus load measurements. The reduction of endpoint severity is reported on an ordinal scale ranging from 0 to 3.

There are more than 59 references containing more than 259 animal model experiments, nearly all involving SARS-CoV-2, SARS-CoV, or MERS-CoV. Approximately 70% of the studies involve mice; 10% involve non-human primates (rhesus macaques, marmosets, and cynomolgus macaques); and 20% involve hamsters, ferrets, cats, dogs, or rabbits. The most commonly studied interventions have included monoclonal antibodies, fusion inhibitors, interferons, and the nucleoside analog polymerase inhibitors.

#### 2.1.4. Clinical Studies

The clinical studies are represented using several enumerated and free-text fields. The enumerated fields include the reference, virus category, and type of study (e.g., observational, randomized trial, randomized placebo-controlled trial). The free-text fields include descriptions of the interventions and regimen details, the study population and methods, and the study findings. CoV-RDB does not provide an assessment of study quality such as validity and risk of bias as there are other research groups providing this type of assessment.

### 2.2. Lookup/Explanation Tables

#### 2.2.1. Virus Categories

Antiviral data on coronaviruses other than SARS-CoV-2 provide insight into the robustness of an antiviral compound, in that, compounds that are active against multiple viral species will be more likely to inhibit future pandemic coronaviruses and will be less vulnerable to the development of drug-resistance mutations. Indeed, for many drug targets, such as the virus RdRP and 3CL^pro^ enzymes, and for host processes upon which coronaviruses depend, inhibitory compounds will likely have broad-spectrum activity.

CoV-RDB contains antiviral data for six categories of coronaviruses: SARS-CoV-2, SARS-CoV, MERS-CoV, endemic human coronaviruses, bat coronaviruses, and non-bat mammalian coronaviruses. SARS-CoV-2 (37%), SARS-CoV (31%), and MERS-CoV (17%) account for 85% of the data. However, the proportion of data associated with SARS-CoV-2 is rapidly increasing. Figure 2 shows the distribution of study types for SARS-CoV-2, SARS-CoV, and MERS-CoV.

SARS-CoV and SARS-CoV-2 belong to the same betacoronavirus 2b (sarbecovirus) clade, and their amino acids are approximately 97% identical in the RdRP and 3CL^pro^ enzymes and 84% identical in the spike protein. In contrast, MERS, a clade 2c betacoronavirus, is approximately 75% identical to SARS-CoV and SARS-CoV-2 in the *RdRP* gene, 60% identical in the *3CL*^pro^ gene, and 40% identical in the *S* gene. Within each of these three viruses, there is little diversity, with median pairwise distances ranging between 0% and 0.2%.

The four endemic human coronaviruses include two clade 2a betacoronaviruses and two alphacoronaviruses. Bat coronaviruses are distributed widely among different clades [2,3]. Indeed, 4 of the 9 betacoronavirus clades and 7 of 11 coronavirus clades are found only in bats. The mammalian coronaviruses include murine hepatitis virus (MHV), which is a longstanding experimental model for coronavirus infection, and several other coronaviruses that have been studied because they are important livestock diseases [4]. Although infectious bronchitis virus is an avian gammacoronavirus, we have included it in the non-bat mammalian coronavirus category.

#### 2.2.2. Virus Isolates/Strains

CoV-RDB uses the terms isolate and strains to describe the different viruses used in antiviral studies, although “strains” is usually reserved for describing isolates that have distinct phenotypic properties [5]. Where possible, isolates are named according to the recommendations from the International Committee on Taxonomy of Viruses [6], i.e., virus/host/location/isolate/date.

Most SARS-CoV-2 isolates are nearly identical to one another with the upper limit for the pairwise amino acid distance being about 0.1%, although this number varies depending upon the gene [7]. Therefore, the biological significance of the isolate used for a particular study is not known. However, for some treatments such as monoclonal antibodies, changes in the sequence encoding the relevant epitope, specifically in the S protein receptor binding domain may prove to have biological and clinical significance [8,9]. Although SARS-CoV resulted from at least two zoonotic introductions from civet cats [4], and although MERS-CoV resulted from multiple zoonotic introductions from dromedary camels, these viruses also demonstrate little genetic variability.

Several of the most commonly used isolates have been cloned, either as intact viruses (e.g., by plaque purification or limiting dilution) or by constructing a cDNA copy representing a single sequence variant. Modification of these clones, such as selection of a resistant variant in vitro [10] or introduction of a reporter gene like GFP or luciferase, presumably retains the characteristics of the original parental virus isolate or strain [11]. Commonly used isolates that have been cloned and manipulated in the laboratory include MERS-CoV/human/Amsterdam/EMC/2012 [12], SARS-CoV/human/Hanoi/Urbani/2003 [13], SARS-CoV-2/human/USA/WA1/2020 [14], and SARS-CoV-2/human/Munich/929/2020 [15].

#### 2.2.3. Cell Lines

The cell lines table provides descriptions for the cell lines used in cell culture and entry assay experiments. It contains four fields: (i) the cell line’s commonly used name, (ii) the source of the cell line, (iii) closely related cell lines, and (iv) a description of the cell line and one or more of the closely related cell lines. The most commonly used cell lines for SARS-CoV and SARS-CoV-2 include a variety of different Vero cell clones [16,17,18,19,20], Huh7 [16,21], Caco-2 [22], Calu-3 [23], and 293T/ACE2 cells [16,17,18] (Table 1). While each of these cell lines expresses ACE2, only Calu-3 cells were originally derived from lung epithelial cells. The 293T cells are typically used for cell–cell fusion and pseudovirus entry experiments. Several studies have also used human alveolar epithelial cells or a variety of different respiratory system or kidney organoids [24,25]. The cell lines used for MERS-CoV are similar, with the main exception that 293T/DPP4 (dipeptidyl peptidase 4) cells are used instead of 293T/ACE2 cells because DPP4 (aka CD26) is the MERS-CoV receptor [18].

#### 2.2.4. Animal Models

The over 10 different animal models used in experiments described in the CoV-RDB include three non-human primate models (rhesus macaques, cynomolgus macaque, and marmosets), multiple transgenic and non-transgenic mouse models, and several additional rodent models including hamsters and ferrets [26,27,28,29,30,31,32,33,34,35,36,37,38,39,40,41,42,43]. The transgenic mice have been modified in multiple ways, including to express hDPP4 so that they can be infected with MERS-CoV, to knock out the interferon (IFN)-α/β receptor to compromise innate immunity [44]; to knock out recombination activating gene 1 (RAG1) to compromise adaptive immunity [45]; to knock out carboxylesterase 1c, which causes poor plasma stability of remdesivir; and to express human rather than mouse ACE2 [39,40,41]. Table 2 describes the utility of the most common non-human primate and mammalian models for studies of the pandemic coronaviruses.

#### 2.2.5. Target and Compound Class

The target table has two main fields: name and description. The target classification organizes drugs, treatments, and compounds according to the virus or host process targeted by a compound including virus enzymes, virus entry into cells, host immunological responses, and other host processes. There are three virus enzyme inhibitor classes: RdRP, protease (including 3CL^pro^, PL^pro^), and helicase inhibitors. There are four inhibitor classes targeting virus entry: convalescent plasma and polyclonal antibody preparations, monoclonal antibodies, miscellaneous other compounds that inhibit virus receptor binding, and fusion inhibitors. There are two classes that target host immunological responses: interferons and other potential immunostimulatory compounds. Although there are potentially many mechanisms by which targeting a host processes may interfere with virus replication, we have divided these into two broad categories: host protease enzymes used by coronaviruses to cleave the spike protein, thus, priming it for fusion and other host proteins or pathways utilized by coronaviruses. The classification of host-acting compounds is likely to continue to evolve as mechanistic pathways become better defined. Table 3 describes each of the targets and compound classes described above. Figure 3 shows the distribution of experimental data types according to target.

#### 2.2.6. Compounds

The database contains experiments involving approximately 1650 compounds. More than 1240 of these compounds appear in the online compounds tables, which contain the following fields: (i) name, (ii) synonyms including abbreviations, (iii) closely related compounds, (iv) drug availability, (v) drug class, (vi) target, and (vii) description. For interferons, the drug class is the type of interferon (α, β, γ, or λ). For monoclonal antibodies, additional data are stored and displayed, including the antibody source and information on sequence and structure availability.

The closely related compounds are subjectively defined as those that we intend to be returned by a query even if that compound was not entered by the user. There are five broad categories of closely related compounds: (i) monoclonal antibodies described in the same publication, (ii) interferons belonging to the same type (i.e., α, β, γ, or λ), (iii) a series of compounds derived from the same lead compound, (iv) prodrugs such as those for GS-441524 (i.e., remdesivir) and β-N-hydroxycytidine (i.e, EIDD-28014), (v) drug combinations such as lopinavir and ritonavir-boosted lopinavir (lopinavir/r), and (vi) compounds presumed to act by a highly similar mechanism of action (e.g., hydroxychloroquine and chloroquine).

The drug availability category indicates whether the compound has been licensed in the U.S. or another country or has been studied in humans. Of 975 compounds with a known or likely mechanism of action, 135 (14%) are U.S. FDA (U.S. Food and Drug Administration) -approved drugs (for indications other than COVID-19), 197 (20%) have been or are currently being evaluated in human clinical trials or are approved outside the U.S., and 595 (61%) are preclinical investigational compounds.

Figure 4 displays EC_50_ values for many of the directly acting antiviral compounds currently in clinical trials for the treatment of COVID-19 including six polymerase inhibitors (remdesivir, EIDD-2801, favipiravir, ribavirin, galidesivir, and sofosbuvir), three HIV-1 protease inhibitors (lopinavir, atazanavir, and darunavir), and three entry inhibitors (receptor binding monoclonal antibodies, soluble recombinant human ACE2, and umifenovir). Figure 5 displays EC_50_ values for many of the repurposed compounds that target host processes required for virus replication including two host PIs that target the transmembrane serine protease 2 (TMPRSS2) enzyme (camostat and nafamostat), three chloroquine analogs that interfere with endosomal acidification (chloroquine, hydroxychloroquine, and mefloquine), three other compounds believed to interfere with membrane trafficking (niclosamide, imatinib, and chlorpromazine), and four compounds acting by a variety of different cellular mechanisms (ivermectin, nitazoxanide, ciclesonide, and cyclosporin).

Figure 4 and Figure 5 show that the potency of currently studied compounds extends over several orders of magnitude with monoclonal antibodies having EC_50_s in the high picomolar to low nanomolar range and some compounds displaying no activity at concentrations above 100 μM. However, there is also marked heterogeneity in the EC_50_ values for the same compound in different experiments. For several drugs, the heterogeneity can be explained by the type of cells used, inoculum size, drug timing, and culture duration. For example, the host TMPRSS2 inhibitors camostat and nafamostat are practically inactive against SARS-CoV-2 in Vero cells but have EC_50_s consistently below 1 μM in Caco-2 and Calu-3 cells, because these cells require TMPRSS2 for virus replication whereas Vero cells do not [22,51,52,53,54,55,56].

Table 4 describes a set of the most promising compounds for the treatment of SARS-CoV-2 based on the following criteria: (i) act by a validated direct or indirect antiviral mechanism, (ii) display sub-micromolar activity in vitro and/or inhibitory activity in an animal model, and (iii) have a record of safety and favorable pharmacokinetics in human subjects. The majority of these compounds are being studied in clinical trials, although the numbers of these trials are far fewer than those for less promising compounds.

### 2.3. Clinical Trials Registry

The Clinical Trials Registry table is a regularly updated, annotated list of ongoing, planned, or completed clinical trials obtained from the ClinicalTrials.gov, WHO ICTRP, and Chinese Clinical Trial websites. It contains trials of compounds with potential antiviral activity but not studies of non-antiviral interventions, such as those designed to optimize intensive-care management or reduce the inflammatory response associated with severe COVID-19. The Clinical Trials Registry classifies trials according to the compound target, the type of trial (e.g., observational or randomized controlled study), the status of the trial (pending, active, or completed), and the population studied. As of 6 August 2020, it contains more than 700 trials of which about 73% are listed on ClinicalTrials.gov and 27% are listed only on the WHO International Clinical Trials Platform.

Figure 6A displays the distribution of planned, ongoing, and published studies according to the compound targets of the drugs studied. Figure 6B displays the same distribution for those drugs in three or more studies. It is notable that many of the most commonly studied compounds have either little or no activity against SARS-CoV-2, including several drugs used for non-coronavirus infections such as the HIV protease inhibitors lopinavir and darunavir and the influenza inhibitors favipiravir, oseltamivir, and umifenovir. The chloroquine analogs, chloroquine and hydroxychloroquine, have weak in vitro activity but have failed to show clinical efficacy in multiple clinical trials [104,105,106,107,108,109,110].

### 2.4. Search Functions

The search function allows users to specify one or more of the following options from four drop-down lists: (i) compound target, (ii) compound, (iii) virus category, and (iv) study type. If the user selects “Any” for one of these and leaves the others in their default position, the search function returns the database’s complete set of cell culture experiments, biochemical experiments, entry assay experiments, animal model studies, and published clinical studies. By selecting one or more of the above options, the search function restricts the data returned to those meeting the search criteria. By using the “Copy to clipboard” link, users can import the results of any query into a spreadsheet where they can further sort and filter query results. The search function also provides a link to the trials in the Clinical Trials registry for selected compounds and compound targets.

The compound drop-down list displays 60 of the most well recognized compounds. Selecting a compound returns the data for that compound as well as for an additional 363 closely related compounds (as described in the compound table Section 2.2.6). If the user selects the compound target from the dropdown menu, then the compound menu will list all compounds designed to inhibit the selected target. The compounds entry on the compounds page also links to all the data on that compound in the database.

## 3. Discussion

To prioritize licensed drugs and investigational compounds for the treatment of COVID-19, it is necessary to compare their relative antiviral activities. Compounds that are not active in vitro will almost certainly not be useful clinically. Therefore, pre-clinical data are necessary to prioritize animal model and clinical studies. Compounds that are active in vitro, however, may also not be clinically useful if their associated in vitro data do not reflect physiologic conditions or if standard dosing with these compounds does not result in sufficient inhibitory concentrations at sites of infection.

The creation of the CoV-RDB was primarily motivated by the observation that many of the drugs being evaluated in CoVID-19 clinical trials demonstrate little or no in vitro anti-coronavirus activity. For example, as recently as July 2020, four of the most commonly studied drugs—chloroquine analogs, azithromycin, lopinavir/r, and favipiravir—demonstrated little if any in vitro activity.

The creation of the CoV-RDB was secondarily motivated by the observation that results for the same compound often vary across different laboratories as a result of experimental design such as cell line, inoculum size, drug-addition timing, duration of culture, and method for measuring virus replication. Given sufficient data, a database makes it possible to eventually identify the experimental features responsible for the heterogeneity in published results, thus improving the ability to compare the antiviral activity of different compounds. Indeed, we have already noted in this manuscript several compound classes for which viral inhibition is influenced by the cells used for virus culture.

The CoV-RDB is also designed to be educational as it provides multiple lookup tables for the viruses, drugs, cell lines, and animal models used in reported experiments. These tables contain descriptions of viruses, virus isolate/strains, cell lines, animal models, and more than 300 licensed and investigational compounds. Work is underway to also add comprehensive, yet detailed, summaries of SARS-CoV-2 monoclonal antibodies and of pharmacokinetic data for those drugs with well-documented in vitro inhibitory activity.

There are several additional web resources devoted to coronavirus drug development including sites devoted to high-throughput drug screening [111], the genetics of monoclonal antibodies [112], and meta-analyses of published clinical trials [113,114]. The National Institutes of Health (NIH) recognizes the importance of such resources and has recently announced a Notice of Special Interest: National Institute of Allergy and Infectious Diseases (NIAID) Priorities for Biomedical Knowledgebases and Repositories (NOT-AI-20-044). The CoV-RDB database, user interface, and underlying computer code represent a framework for organizing a vast amount of data and for facilitating data curation. However, the value of this resource depends upon ongoing manual data curation and annotation.

In conclusion, the CoV-RDB provides a uniquely integrated interdisciplinary synthesis of in vitro, animal model, and clinical studies of compounds with proven or possible anti-coronavirus activity. It helps researchers place their findings in the context of previously published data and it facilitates comparisons between different candidate antiviral compounds, thereby helping scientists, clinical investigators, public health officials, and funding agencies to prioritize the most promising compounds and repurposed drugs for further development.

## Figures and Tables

**Figure 1 viruses-12-01006-f001:**
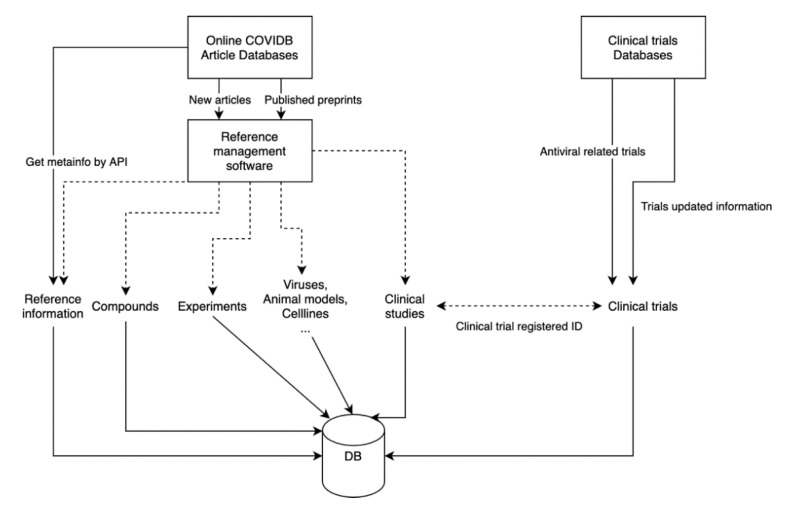
Experiments and clinial trials are semi-automatically downloaded and processed, data are periodically updated.

**Figure 2 viruses-12-01006-f002:**
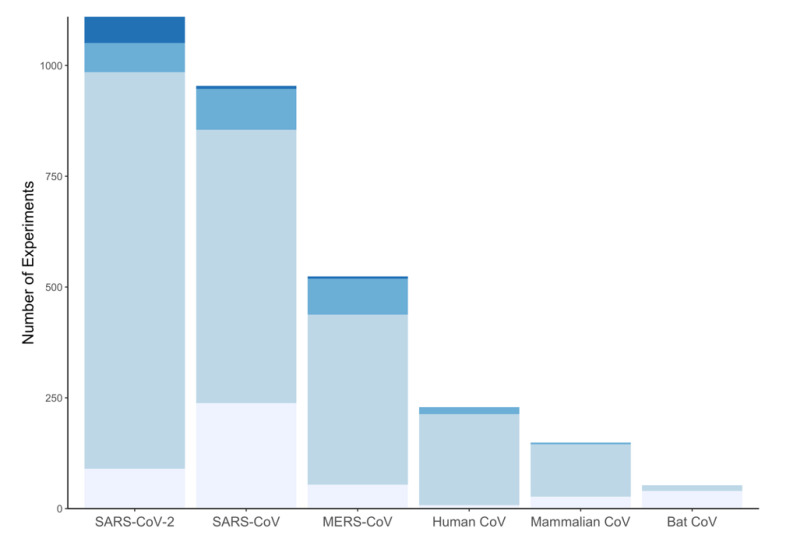
The distribution of biochemical experiments (lightest), cell culture experiments (light), animal model experiments (dark), and clinical studies (darkest) for the six categories of virus in the Coronavirus Antiviral Research Database (CoV-RDB). The cell culture experiments also include entry assay experiments.

**Figure 3 viruses-12-01006-f003:**
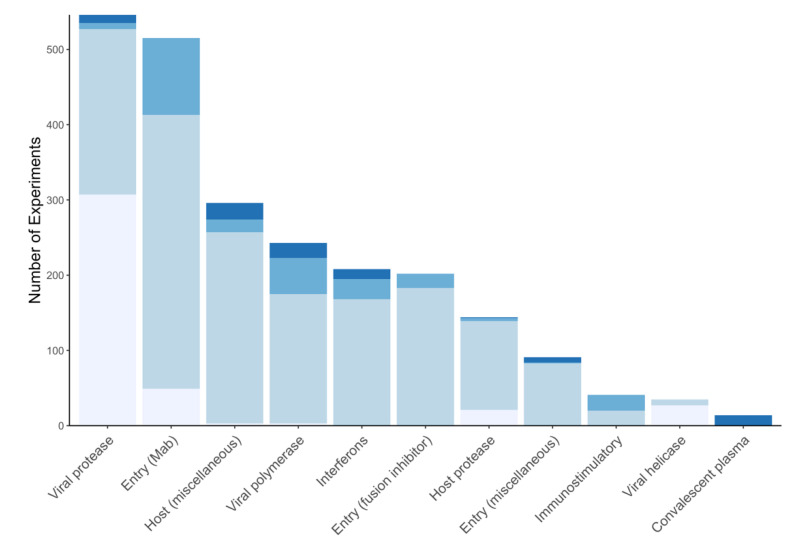
The distribution of biochemical experiments (lightest), cell culture experiments (light), animal model studies (dark), and clinical studies (darkest) for the different targets of antiviral therapy in the Coronavirus Antiviral Research Database (CoV-RDB). The cell culture experiments also include entry assay experiments. The results for approximately 600 experiments involving compounds with an unknown or uncertain mechanism of action are not shown.

**Figure 4 viruses-12-01006-f004:**
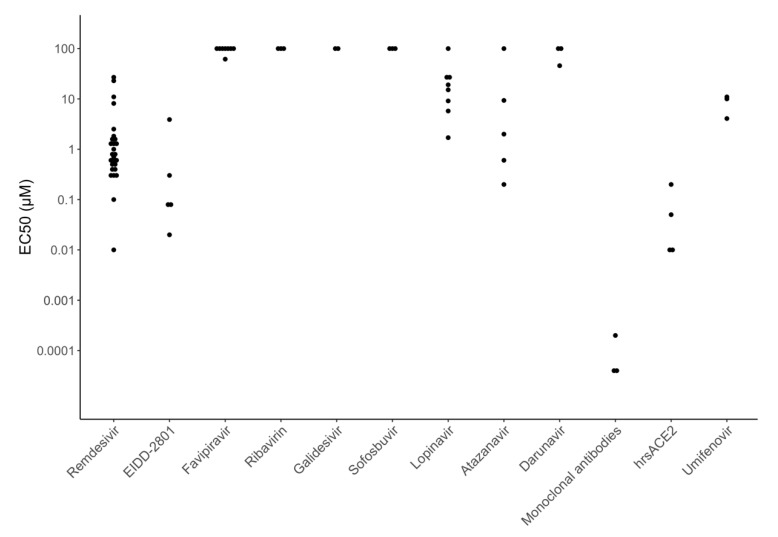
Half maximal effective concentration (EC_50_) values for many of the directly acting antiviral compounds in clinical trials including six polymerase inhibitors (remdesivir, EIDD-2801, favipiravir, ribavirin, galidesivir, and sofosbuvir), three protease inhibitors (lopinavir, atazanavir, and darunavir), and three entry inhibitors (receptor binding monoclonal antibodies, soluble recombinant human angiotensin converting enzyme 2 (ACE2), and umifenovir). EC_50_ values above 100 μM are plotted at 100 μM.

**Figure 5 viruses-12-01006-f005:**
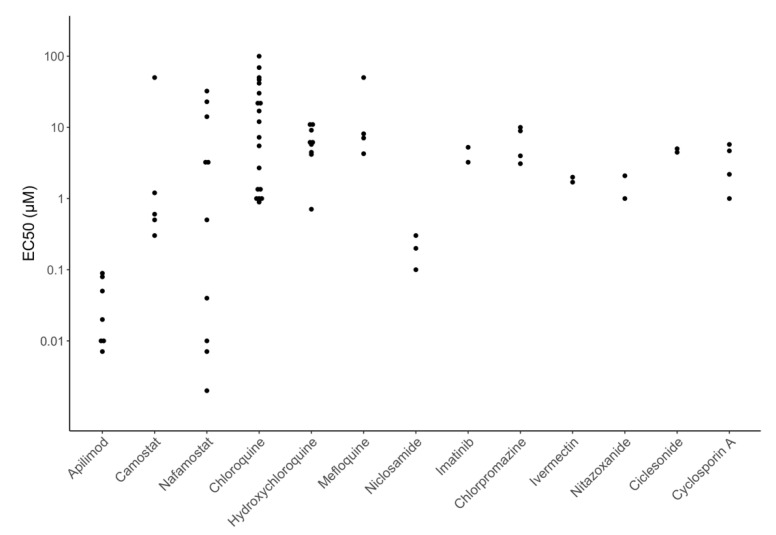
EC_50_ values for many of the repurposed host-acting compounds currently in clinical trials including the host protease inhibitors (camostat and nafamostat), six possible endosomal trafficking inhibitors (chloroquine, hydroxychloroquine, mefloquine, niclosamide, imatinib, chlorpromazine) and four inhibitors acting by a variety of different mechanisms (ivermectin, nitazoxanide, ciclesonide, and cyclosporin).

**Figure 6 viruses-12-01006-f006:**
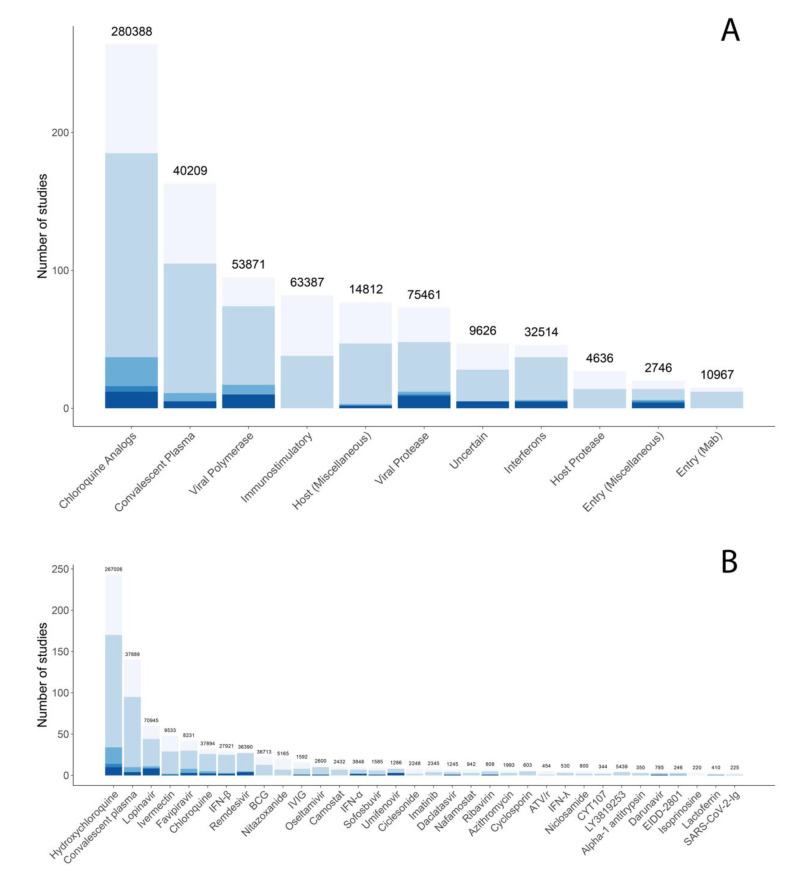
Distribution of targets (**A**) and the most commonly studied compounds (**B**) for published (bottom), ongoing (middle), and planned (top) antiviral clinical trials through August 6. Although chloroquine analogs are considered to act primarily through the inhibition of virus endosomal trafficking, they are separated out from other endosomal trafficking inhibitors in (**A**). (**B**) shows compounds included in three or more trials.

**Table 1 viruses-12-01006-t001:** Frequently used cells for culturing pandemic coronavirus antiviral research.

Cell Line	Source	Coronaviruses	Description
Vero cells (Vero E6, other clones; Vero E6/TMPRSS2)	African green monkey kidney epithelial cell line	MERS-CoV SARS-CoV SARS-CoV-2	Vero cells support the replication of many viruses often producing a visual cytopathic effect [16,17,18]. They express ACE2, the receptor for SARS-CoV and SARS-CoV2, and DPP4, the receptor for MERS-CoV. Although Vero cells are IFN-deficient, they express the IFN-α/β receptor and thus retain the ability to respond to exogenous IFN [19]. Vero E6 cells engineered to express greater amounts of TMPRSS2 produce higher SARS-CoV-2 titers of SARS-CoV-2 [20]. Drugs that target TMPRSS2 are often inactive in Vero cells.
Calu-3 2B4	Human lung epithelial cell line	MERS-CoV SARS-CoV SARS-CoV-2	Calu-3 cells form differentiated pseudostratified columnar epithelia highly permissible to coronavirus infection. They are polarized with an apical domain facing the airway lumen and a basolateral domain facing internally. They produce a visual cytopathic effect. The 2B4 clone has high ACE2 expression. They are often used for the preclinical development of respiratory drugs [23].
CaCo-2	Heterogeneous human epithelial colorectal adenocarcinoma	SARS-CoV SARS-CoV-2	CaCo-2 cells are considered to be more pharmacologically relevant than Vero cells for some studies because of their human origin [22].
Huh-7	Human hepatoma	MERS-CoV SARS-CoV SARS-CoV-2	Huh-7 cells express ACE2 and TMPRSS2, yet do not support levels of replication as high as Vero cells [16,21].
HEK-293T/ACE2 (HEK-293T/DPP4)	Human embryonic kidney	MERS-CoV SARS-CoV SARS-CoV-2	The 293T cells are derived from the human embryonic kidney 293 cell line. 293T cells contain the SV40 large T-antigen, which facilitates replication of transfected plasmids containing the SV40 origin of replication. The 293T/ACE2 cells are transfected to express ACE2 and have been used for many SARS-CoV cell–cell fusion and pseudovirus entry inhibitor studies [16,17,18].
HAE	Human airway epithelial cells	MERS-CoV SARS-CoV SARS-CoV-2	Differentiated human airway cells have occasionally been used to study antiviral agents, although they are more commonly used to study viral pathogenesis [24,25].

TMPRSS2—transmembrane serine protease 2; ACE2—angiotensin converting enzyme 2; IFN—interferon; DPP4—dipeptidyl peptidase 4; SV40—simian virus 40.

**Table 2 viruses-12-01006-t002:** Animal models for pandemic coronavirus antiviral research.

Species	Coronaviruses Used	Comments
Mouse (C57BL/6, Balb/c)	MERS-CoV SARS-CoV SARS-CoV-2	Pathological changes observed in the aged mouse model infected with SARS-CoV more closely resemble those observed in humans [29]. *RAG* −/− mice lack T and B cells and lack adaptive immunity and experience prolonged coronavirus shedding [45]. *IFNAR* −/− mice are vulnerable to greater coronavirus disease severity [44].
Transgenic *hACE2* mice	SARS-CoV SARS-CoV-2	There are many *hACE2* transgenic mouse models. These mice are more likely to experience weight loss, detectable virus loads, and interstitial pneumonia following challenge with SARS-CoV-2 than those with the murine ACE2 receptor [39,40,41].
Rhesus Macaque	MERS-CoV SARS-CoV SARS-CoV-2	Infection causes a self-limiting disease associated with virus replication. Radiographic and pathologic examination of SARS-CoV-2-infected animals display evidence of pneumonia [26,30,32,42].
Cynomolgus Macaque	MERS-CoV SARS-CoV SARS-CoV-2	Infection results in a productive infection in respiratory epithelial cells. Symptoms are minimal but virus shedding can last up to 2 weeks. Chest radiographs reveal unifocal or multifocal pneumonia. Autopsy reveals variable amounts of foci of diffuse alveolar damage [31,33].
Common Marmoset	SARS-CoV MERS-CoV	Infection causes severe acute disease that mimics severe human infection [27,46].
Ferret	SARS-CoV SARS-CoV-2	Upon infection, ferrets develop fevers and shed viruses in their upper airways, urine, and feces for up to 8 days. They can also transmit the infection to other ferrets [35,36,37].
Syrian hamster	SARS-CoV SARS-CoV-2	SARS-CoV and SARS-CoV-2, but not MERS-CoV, cause a self-limited respiratory tract infection in hamsters. Infection is associated with high-levels of virus and areas of lung pathology [28,34,38,43].

*RAG*—recombination activating gene; *IFNAR*—interferon-α/β receptor; *hACE2*—human angiotensin-converting enzyme 2.

**Table 3 viruses-12-01006-t003:** Antiviral coronavirus therapy targets.

Target	Compound Class	Description of Target/Compound Class
Virus enzymes	Polymerase inhibitors	Inhibitors of the coronavirus RNA-directed RNA polymerase (RdRP) enzymes include nucleoside analogs that cause immediate chain termination, delayed chain termination, or viral mutagenesis.
	Protease inhibitors	Coronaviruses contain two protease enzymes: 3 chymotrypsin-like cysteine protease (3CL^pro^ or Main (M)-pro) and papain-like (PL^pro^).
	Helicase inhibitors	Coronavirus helicases catalyze the unwinding of duplex RNA molecules into single strands.
Entry	Convalescent plasma and polyclonal sera.	Convalescent plasma is one of the most widely studied treatments for COVID-19. Polyclonal sera and immunoglobuline preparations have also entered clinical trials.
	Monoclonal antibodies	Recovery during SARS-CoV, MERS-CoV, and SARS-CoV-2 is usually associated with the development of neutralizing antibodies. Most SARS-CoV-2 neutralizing mAbs target the part of the receptor binding domain (RBD) that binds ACE2 while most MERS-CoV neutralizing mAbs target the part of the RBD that binds DPP4. Many highly potent neutralizing mAbs targeting each of the pandemic coronaviruses have shown protection in vitro and in animal models. Structural studies have defined specific RBD epitopes recognized by individual mAbs and identified amino acid residues that are critical for mAb binding.
	Other receptor binding inhibitors	SARS-CoV and SARS-CoV-2 spike S1 binds to the cellular angiotensin converting enzyme 2 (ACE2) receptor. MERS-CoV binds to dipeptidyl peptidase 4 (DPP4). A variety of compounds including non-antibody proteins, peptides, and small molecules have been shown to prevent the binding of the coronavirus spike protein to its cellular receptor.
	Fusion inhibitors	Following receptor binding and spike S1/S2 cleavage and S2 priming, heptad region 1 (HR1), which is close to the fusion peptide sequence, and HR2, which is close to the virus membrane, collapse on to one another to bring virus and cell membranes together. Nearly all fusion inhibitors are HR2-mimicking peptides less than 70 kDa that bind HR1, thus preventing HR1−HR2 binding.
Immunologic processes	Interferons	Interferons have been extensively studied for their ability to inhibit each of the pandemic coronaviruses in cell culture, animal models, and/or clinical studies [47]. SARS-CoV-2 may be more susceptible to interferons than SARS-CoV is [48].
	Immunostimulatory compounds	There are several clinical trials using immunostimulatory cytokines and compounds purported to induce interferon.
Host processes	Host protease inhibitors	Cleavage of coronavirus spike proteins is necessary for the virus to transition from receptor attachment to cell fusion. For SARS-CoV-2, there is a poly-basic furin cleavage site at the S1/S2 boundary and another cleavage site within S2 believed to be cleaved at the cell surface by host TMPRSS2 enzymes [49].
	Miscellaneous	Multiple intracellular processes essential to virus replication are vulnerable to pharmacologic inhibitors including endosomal acidification, membrane formation, various signaling pathways, nucleotide biosynthesis, and autophagy [50].
Uncertain	Miscellaneous	Many compounds with uncertain mechanisms of action have been found to inhibit coronaviruses in vitro. Several of these are also being studied in clinical trials.

mAbs—monoclonal antibodies.

**Table 4 viruses-12-01006-t004:** Promising SARS-CoV-2 antiviral compounds.

Compound Class	Compound	Description
Polymerase inhibitors	Remdesivir	Remdesivir is a delayed chain terminator monophosphate prodrug of a 1′-cyano-substituted adenine C-nucleoside analog. It has high nanomolar inhibitory activity in vitro against SARS-CoV-2 particularly in cells other than Vero cells [21,22,57,58,59]. It reduces viral replication and lung pathology in mice and rhesus macaques when administered shortly after infection [58,60]. In a double-blind randomized clinical trial, its intravenous administration led to a significant reduction in time to recovery from 15 to 11 days (*p* < 0.001) and a non-statistically significant reduction in day 14 mortality of 11.9% vs. 7.1% (*p* = 0.06) [61]. Based on this trial, the FDA issued an emergency use authorization for remdesivir in patients with severe COVID-19. Ongoing trials are examining its safety and efficacy when administered subcutaneously or via inhalation.
	β-D-N4-hydroxycytidine-5′-isopropyl ester (EIDD-2801)	EIDD-2801 is a nucleoside analog, which like remdesivir has high nanomolar inhibitory activity in vitro against SARS-CoV-2 [62]. It reduces SARS-CoV and MERS-CoV replication and lung pathology in a mouse model [62]. It is being evaluated in two phase II clinical trials.
Monoclonal antibodies (mAbs)	REGN10933 + REGN10987 (phase III trials)	REGN10933 and REGN10987 are mAbs with subnanomolar inhibitory activity that bind to non-overlapping ACE2-competing SARS-CoV-2 spike receptor binding domain epitopes [8,63]. This mAb combination also reduces virus replication and lung pathology in Syrian hamsters and rhesus macaques [64]. The combination is being evaluated in phase III trials for preventing and treating SARS-CoV-2 infection.
	LY3819253 (phase III trials)	LY3819253 is a SARS-CoV-2 mAb in phase III trials for preventing and treating COVID-19. As of August 2020, there are no associated published preclinical data.
	mAbs (phase I/II trials)	AZD7442, BRII-196, JS016, SCTA01, STI-1499, and TY027 are mAbs in phase I/II trials. As of August 2020, there are no associated published preclinical data linked to mAbs with these names.
	mAbs (preclinical)	Many research groups that have published preclinical data on one or more mAbs (or mAb variants such as nanobodies) including their in vitro inhibitory activity, genetic sequence data, three-dimensional structural data, and/or animal model data [65,66,67,68,69,70,71,72,73,74,75,76,77,78,79,80].
Interferons	IFN-α, IFN-β, and IFN-λ	IFN-α, IFN-β, and IFN-λ each inhibit SARS-CoV-2 by 90%–99% at low concentrations of about 100 international units (IU)/mL [48,81,82,83,84]. Inhalational IFN-α and parenteral IFN-β were associated with modest reductions in disease severity and/or virus loads in two small open-label clinical trials [84,85]. An inhaled formulation of IFN-β was reported in the news to significantly reduce the odds of developing severe disease or death in a blinded randomized control trial (SNG016) of 220 patients that has not yet been published (https://www.synairgen.com/covid-19/). There are currently four planned or ongoing placebo-controlled trials of parenteral or inhaled IFN-β (~1800 patients) and of parenteral IFN-λ (~400 patients).
Host protease inhibitors	Camostat and nafamostat	Camostat and nafamostat are TMPRSS2 inhibitors with nanomolar coronavirus inhibitory activity in biochemical and cell culture assays [51,53,54,55,56,86,87,88]. Both drugs are used in Japan for the treatment of pancreatitis, while nafamostat is also used as an anticoagulant and for the treatment of disseminated intravascular coagulation. Although nafamostat has approximately 10-fold greater inhibitory activity than camostat, it may be associated with greater toxicity. Camostat is being studied in two blinded and two open-label randomized controlled studies totaling about 900 patients. Nafamostat is being studied in three small randomized open-label studies totaling about 200 patients.
Host miscellaneous	Apilimod	Apilimod was found to inhibit SARS-CoV-2 at two-digit nanomolar levels with high selectivity indexes in multiple drug screens [89,90,91,92]. It inhibits the membrane protein PI(3,5)P2 by inhibiting the enzyme PI-3P-5-kinase (PIKfyve) thus interfering with endosomal trafficking of SARS-CoV-2 and additional viruses utilizing the same endosomal pathway [93,94]. It has been studied in humans in multiple clinical trials and been found to be safe and well tolerated. It is being studied for the treatment of mild SARS-CoV-2 infections in one randomized placebo-controlled phase II trial.
	PTC299	PTC299 is an inhibitor of dihydroorotate dehydrogenase (DHODH), a rate limiting enzyme in the pyrimidine biosynthesis pathway [95]. DHODH inhibitors are therapeutic targets for autoimmune disases and viral infections [96,97]. PTC299 has been found to be safe and have favorable pharmacokinetics in more than 300 human subjects and has low nanomolar SARS-CoV-2 inhibitory activity and a high selectivity index [98]. As both viral replication and cytokine overproduction depend on pyrmidine synthesis, DHODH inhibition may have a dual role in COVID-19 treatment. DHODH inhibition is synergistic with viral polymerase inhibition [97]. There is one phase II/III trial of PTC299 for patients with severe but not critical COVID-19. Leflunomide, another repurposed DHODH inhibitor was found to reduce virus load in an open-label pilot study of 27 patients [99].
Receptor binding	Soluble recombinant human ACE2 (rhACE2)	rhACE2 protects mice for SARS-CoV-1 ARDS and has been studied as a treatment for ARDS in humans [100,101]. It inhibits SARS-CoV-2 spike binding at nanomolar concentrations in a wide variety of cell lines [102,103]. There are two ongoing phase II trials of an intravenous commercial rhACE2 preparation.

Notes: Some compounds with sub-micromolar in vitro activity are not included in this table either because (i) they have not been used in humans, (ii) they have been identified only in high-throughput drug screens, or (iii) they have unfavorable pharmacokinetics. Several compounds that inhibit SARS-CoV-2 less potently but are being studied as inhalational and/or intranasal therapies are also not shown. This last category includes (i) compounds that inhibit the interaction of SARS-CoV-2 spike and cell surface heparin sulfate proteoglycans such as lactoferrin and heparin; and (ii) ciclesonide, an inhaled corticosteroid that may interfere with membrane trafficking by binding directly to nsp-3 or nsp-4 or indirectly through a host protein.

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
