# Peer review of "Coronavirus Antiviral Research Database (CoV-RDB): An Online Database Designed to Facilitate Comparisons between Candidate Anti-Coronavirus Compounds"

_viruses, 2020, doi:10.3390/v12091006_

Round 1
Reviewer 1 Report
The manuscript by Tzou et al. entitled “Coronavirus Antiviral Research Database (CoV-RDB): An Online Database Designed to Facilitate Comparisons Between Candidate Anti-Coronavirus Compounds” reports the development of an online relational database that compiles data about experiments exploring antiviral effects of compounds against different members of the Coronaviridae family of viruses using cell culture, animal models and clinical studies reported in the scientific literature and in clinical trial databases (e.g. Clinicaltrials.gov). The authors have put in considerable effort to compile the information buried in the scientific literature about antiviral effects of candidate drugs and representing this information in a standardized format that facilitates reliable search and exploration. They appear to have captured the most salient information of relevance for the different study types. In general, it appears that this would be a useful resource for researcher interested in coronavirus antivirals.
Major comments:
- The biggest concern relates to the sustainability of the resource. Since the content of the database is derived from the manual curation of the scientific literature, how do the authors plan on keeping it up to date. A maintenance plan for keeping the content up to date should be described. It also appears that the funding used was from the R24AI136618 grant for development of an HIV Drug Resistance Database. Without any dedicated funding for the maintenance and further development of this database, it is unclear if the community can depend on its availability in the future. How do the authors plan on supporting the resource moving foreward?
- In terms of the manuscript, in addition to describing the resource, the authors should also provide a more detailed review of the data content they have collected from the literature to include more of an analysis of which compounds appear to be effective on which viruses under which experimental conditions. They have put substantial effort extracting this information from the literature. They should also provide some level of interpretation in the manuscript to add additional value.
Minor comments:
- Not all facets collected are available for filtering. For example, users might want to limit animal studies to those only involving non-human primate hosts, but this facet is not available for filtering.
- In some cases, different data are returned in different sections – search for study type “clinical study” returns different data from the “clinical trials” link at top of the home page. It’s not clear what the difference is.
- From the Clinical Studies table, “Copy to clipboard” and pasting into a spreadsheet does not produce one row per record; the author/year is in a separate row from the journal.
- Monoclonal antibodies and convalescent plasma are listed as targets, but these are actually therapeutic agents that target virus structural proteins. This should be corrected in their semantic model.
- The authors need to describe how “closely related compounds” are defined.
- The animal models table has symbols that are not explained (e.g. <=>).
Author Response
Major comments:
- The biggest concern relates to the sustainability of the resource. Since the content of the database is derived from the manual curation of the scientific literature, how do the authors plan on keeping it up to date. A maintenance plan for keeping the content up to date should be described. It also appears that the funding used was from the R24AI136618 grant for development of an HIV Drug Resistance Database. Without any dedicated funding for the maintenance and further development of this database, it is unclear if the community can depend on its availability in the future. How do the authors plan on supporting the resource moving forward? The reviewer raises a critical point which we now address in the next-to-last paragraph of the discussion. The italicized sentences are new: “There are several additional web resources devoted to coronavirus drug development including sites devoted to high-throughput drug screening (Brimacombe et al. 2020), the genetics of monoclonal antibodies (Raybould et al. 2020), and meta-analyses of published clinical trials (Yang et al. 2020; Lythgoe and Middleton 2020). The NIH recognizes the importance of such resources and has recently announced a Notice of Special Interest: NIAID Priorities for Biomedical Knowledgebases & Repositories (NOT-AI-20-044). The CoV-RDB database, user interface, and underlying computer code represent a framework for organizing a vast amount of data and for facilitating data curation. However, the value of this resource depends upon ongoing manual data curation and annotation.”
- In terms of the manuscript, in addition to describing the resource, the authors should also provide a more detailed review of the data content they have collected from the literature to include more of an analysis of which compounds appear to be effective on which viruses under which experimental conditions. They have put substantial effort extracting this information from the literature. They should also provide some level of interpretation in the manuscript to add additional value. We have added Table 4, which provides an integrated summary of promising antiviral compounds. Table 4 is described in the section on Compounds (2.2.4; lines 289-294): “Table 4 describes a set of the most promising compounds for the treatment of SARS-CoV-2 based on the following criteria: (i) act by a validated direct or indirect antiviral mechanism; (ii) display sub-micromolar activity in vitro and/or inhibitory activity in an animal model; and (iii) have a record of safety and favorable pharmacokinetics in human subjects. The majority of these compounds are being studied in clinical trials, although the numbers of these trials are far fewer than those for less promising compounds.”
Minor comments:
- Not all facets collected are available for filtering. For example, users might want to limit animal studies to those only involving non-human primate hosts, but this facet is not available for filtering. We have added the following sentence to the first paragraph in the section 2.4 Search functions (lines 293-294): “By using the “Copy to clipboard” link, users can import the results of any query into a spreadsheet where they can further sort and filter query results.”
- In some cases, different data are returned in different sections – search for study type “clinical study” returns different data from the “clinical trials” link at top of the home page. It’s not clear what the difference is. To clarify this, we changed “Clinical Studies” to “Published Clinical Studies” and “Clinical Trials” to “Clinical Trials Registry”
- From the Clinical Studies table, “Copy to clipboard” and pasting into a spreadsheet does not produce one row per record; the author/year is in a separate row from the journal. We have corrected this error.
- Monoclonal antibodies and convalescent plasma are listed as targets, but these are actually therapeutic agents that target virus structural proteins. This should be corrected in their semantic model. We have updated the target names as follows. The four classes of compounds that target virus entry have been renamed as follows: Entry (monoclonal antibodies), Entry (convalescent plasma), Entry (fusion inhibitor), and Entry (miscellaneous). We have also modified the classification for host-acting drug because it can be difficult to determine the precise mechanism of host-acting drugs that are not protease inhibitors. Therefore, we have combined those drugs in the host endosome and host miscellaneous categories into a single category called host miscellaneous. However, we note (lines 223-224) that “The classification of host-acting compounds is likely to continue to evolve as mechanistic pathways become better defined.”
- The authors need to describe how “closely related compounds” are defined. We have expanded the explanation for “closely related compounds” and it is now the second paragraph of the drug section (lines 208-216): “The closely related compounds are subjectively defined as those that we intend to be returned by a query even if that compound was not entered by the user. There are five broad categories of closely related compounds: (i) monoclonal antibodies described in the same publication; (ii) interferons belonging to the same type (i.e., α, β, γ, or λ); (iii) a series of compounds derived from the same lead compound; (iv) prodrugs such as those for GS-441524 (i.e., remdesivir) and β-N-hydroxycytidine (i.e, EIDD-28014); (v) drug combinations such as lopinavir and ritonavir-boosted lopinavir (lopinavir/r); and (vi) compounds presumed to act by a highly similar mechanism of action (e.g., hydroxychloroquine and chloroquine).”
- The animal models table has symbols that are not explained (e.g. <=>). In the header, we have added the information iconⓘ, which upon mouse over shows the following: “↓, ↓↓↓ decreased; ↑, ↑↑↑ increased; <=> no discernable difference; ND not done; ? could not determine”
Reviewer 2 Report
A nice review dedicated to a new online database on drugs preclinically and sometimes clinically tested on coronavirus (and others). Following the description, the database is easy to use. I just regret the drugs are not referenced using their CAS number that should be more easier to compare biological activities.
It should be of interest the authors conclude with the last FDA and/or EMA approvals
Author Response
A nice review dedicated to a new online database on drugs preclinically and sometimes clinically tested on coronavirus (and others). Following the description, the database is easy to use. I just regret the drugs are not referenced using their CAS number that should be easier to compare biological activities. We searched the PubChem compound database for each of the compounds and their synonyms in the compounds table. This enabled us to link 255 compounds to their CAS numbers.
It should be of interest the authors conclude with the last FDA and/or EMA approvals. In response to the first reviewer, we have added a table (Table 4) that describes promising SARS-CoV-2 antiviral compounds as well as the current status of their clinical trials. There are currently no drugs that have been approved by the FDA or EMA. However, we have noted in Table 4 that remdesivir was awarded an emergency use authorization based on the results of a placebo-controlled randomized trial.
Corresponding response to reviewer 1:
- In terms of the manuscript, in addition to describing the resource, the authors should also provide a more detailed review of the data content they have collected from the literature to include more of an analysis of which compounds appear to be effective on which viruses under which experimental conditions. They have put substantial effort extracting this information from the literature. They should also provide some level of interpretation in the manuscript to add additional value. We have added Table 4, which provides an integrated summary of promising antiviral compounds. Table 4 is described in the section on Compounds (2.2.4; lines 289-294): “Table 4 describes a set of the most promising compounds for the treatment of SARS-CoV-2 based on the following criteria: (i) act by a validated direct or indirect antiviral mechanism; (ii) display sub-micromolar activity in vitro and/or inhibitory activity in an animal model; and (iii) have a record of safety and favorable pharmacokinetics in human subjects. The majority of these compounds are being studied in clinical trials, although the numbers of these trials are far fewer than those for less promising compounds.”